# STOCHASTIC PROTOTYPE EMBEDDINGS

## ABSTRACT

Supervised deep-embedding methods project inputs of a domain to a representational space in which same-class instances lie near one another and different-class instances lie far apart. We propose a probabilistic method that treats embeddings as random variables. Extending a state-of-the-art deterministic method, Prototypical Networks (Snell et al., 2017), our approach supposes the existence of a class prototype around which class instances are Gaussian distributed. The prototype posterior is a product distribution over labeled instances, and query instances are classified by marginalizing relative prototype proximity over embedding uncertainty. We describe an efficient sampler for approximate inference that allows us to train the model at roughly the same space and time cost as its deterministic sibling. Incorporating uncertainty improves performance on few-shot learning and gracefully handles label noise and out-of-distribution inputs. Compared to the state-of-the-art stochastic method, Hedged Instance Embeddings (Oh et al., 2019), we achieve superior large- and open-set classification accuracy. Our method also aligns class-discriminating features with the axes of the embedding space, yielding an interpretable, disentangled representation.

## 1 INTRODUCTION

Supervised deep-embedding methods map instances from an input space to a latent embedding space in which same-label pairs are near and different-label pairs are far. The embedding thus captures semantic relationships without discarding inter-class structure. In contrast, consider a standard neural network classifier with a softmax output layer trained with a cross-entropy loss. Although its penultimate layer might be treated as an embedding, the classifier's training objective attempts to orthogonalize all classes and thereby eliminate any information about inter-class structure. Supervised embedding methods are critical for large- and open-set classification tasks, and are popular for few-shot and lifelong learning tasks.

Nearly all previous methods for deep embeddings are deterministic: an instance projects to a point in the embedding space. Deterministic embeddings fail to capture uncertainty due either to out-of-distribution inputs (e.g., data corruption) or label ambiguity (e.g., overlapping classes). Representing uncertainty is important for many reasons, including robust classification and decision making, informing downstream models, interpreting representations, and detecting out-of-distribution samples. In this article, we propose a method for discovering *stochastic* embeddings, where each embedded instance is a random variable whose distribution reflects the uncertainty in the embedding space.

Our proposed method, the *Stochastic Prototype Embedding (SPE)*, is an extension of the *Prototypical Network (PN)* (Snell et al., 2017). As in the PN, our SPE assumes each class can be characterized by a prototype in the embedding space and an instance is classified based on its proximity to a prototype. In the case of the SPE, the embeddings and prototypes are Gaussian random variables, each class instance is assumed to be a Gaussian perturbation of the prototype, and a query instance is classified by marginalizing over the embedding uncertainty. Our main contribution is to show that SPE outperforms the only other fully-formulated method for stochastic supervised embeddings, the *Hedged Instance Embedding (HIB)* (Oh et al., 2019), on a superset of the complete battery of experiments used to justify HIB. SPE is also more computation efficient to train than HIB, with complexity comparable to that of the PN, and has no hand-tuned parameters. We also demonstrate that embedding distributions are related to label uncertainty and input ambiguity. Finally, we explore an intriguing emergent property of SPE: that it attains more interpretable representations by disentangling class-discriminative features.

## 2 RELATED WORK

Supervised embedding methods are popular in the few-shot learning literature (Koch et al., 2015; Vinyals et al., 2016; Snell et al., 2017; Triantafillou et al., 2017; Finn et al., 2017; Edwards and Storkey, 2017; Scott et al., 2018; Ridgeway and Mozer, 2018; Mishra et al., 2018) where the goal is to classify query instances based on one or a small number of labeled exemplars of novel classes. These methods operate by embedding the queries and exemplars using a pre-trained network, and classifying each query according to its proximity to the exemplars. Embedding methods are also critical in open-set recognition domains such as face recognition and person re-identification (Chopra et al., 2005; Li et al., 2014; Yi et al., 2014; Zheng et al., 2015; Schroff et al., 2015; Liu et al., 2015; Ustinova and Lempitsky, 2016; Song et al., 2016; Wang et al., 2017).

Loss functions used to obtain embeddings can be characterized according to the number of instances required to specify a loss. To describe these losses, we will use the notation $z_\alpha$ for an embedding of class $\alpha$. *Pairwise* losses attempt to minimize within-class distances, $||z_\alpha - z'_\alpha||$, and maximize between-class distances, $||z_\alpha - z_\beta||$ (Chopra et al., 2005; Hadsell et al., 2006; Yi et al., 2014). *Triplet* losses attempt to ensure within-class instances are closer than between-class instances, $||z_\alpha - z'_\alpha|| < ||z_\alpha - z_\beta||$ (Schroff et al., 2015; Song et al., 2016; Wang et al., 2017). *Quadruplet* losses attempt to ensure every within-class pair is closer than every between-class pair, $||z_\alpha - z'_\alpha|| < ||z''_\alpha - z_\beta||$ (Ustinova and Lempitsky, 2016). Finally, *cluster-based losses* attempt to use all instances of a class (Rippel et al., 2016; Fort, 2017; Song et al., 2017; Snell et al., 2017; Ridgeway and Mozer, 2018). In particular, the Prototypical Network (Snell et al., 2017) computes the mean of a set of instances of a class, $\bar{z}_\alpha$, and ensures that additional instances of that class, $z_\alpha$, satisfy a proximity constraint such as $||z_\alpha - \bar{z}_\alpha|| < ||z_\alpha - \bar{z}_\beta||$. Cluster-based methods represent state-of-the-art over, in particular, pairwise and triplet losses, as one might expect given the chronology of publications.

Probabilistic embedding methods have recently appeared. Allen et al. (2019) extend PNs via Bayesian nonparametrics to treat each prototype as a mixture distribution, though they do not allow uncertainty in the embedding space, the critical element of our work. Vilnis and McCallum (2018) propose a method for learning density-based word embeddings, but it is *unsupervised*. Deep Variational Transfer (Belhaj et al., 2018) is a generative form of the discriminative model we propose, which requires modeling input distributions; this work tackled the somewhat different problem of covariate shift. The method closest to ours is an extension of PNs that uses a Mahalanobis distance instead of a Euclidean distance to assess similarity (Fort, 2017). Although this method lacks probabilistic semantics, it has similarity with SPE (comparison in Appendix G).

Two prior methods have been proposed for discovering stochastic embeddings in a supervised setting, i.e., for few-shot and open-set recognition. The *Hedged Instance Embedding (HIB)* (Oh et al., 2019) utilizes a probabilistic alternative to the contrastive loss and is trained using a variational approximation to the information bottleneck principle. HIB is critically dependent on a constant, $\beta$, that determines characteristics of an information bottleneck (i.e., how much of the input entropy is retained in the embedding). Choosing this constant is a matter of art. The *Oracle-Prioritized Belief Network (OPBN)* (Karaletsos et al., 2016) is a generative model that learns a joint distribution over inputs and oracle-provided triplet constraints. The OPBN was not tested on few-shot and open-set recognition because it requires extensions to be applied to classification tasks. In the deterministic setting, Scott et al. (2018) argue that cluster-based methods outperform pairwise and triplet methods; thus, we have reason to expect that in a stochastic setting, a cluster-based method like the one we propose in this article, SPE, will outperform pairwise (HIB) and triplet (OPBN) methods.

## 3 THE MODEL

The SPE assumes that the latent representation, $z$, is a Gaussian RV conditioned on the input, $x$:

$$p(z|x) = \mathcal{N}(z; \mu_x, \sigma_x^2 I) \qquad (1)$$

with mean, $\mu_x$, and variance, $\sigma_x^2$, computed by a deep neural network, similar to a Variational Autoencoder (Kingma and Welling, 2014). The classification, $y$, in turn is conditioned on $z$, with $p(y|z)$ taking the same form as in the original PN (Snell et al., 2017), to be described shortly. Given an input, a class prediction is made by marginalizing over the embedding uncertainty:

$$p(y|x) = \int_z p(y|z)p(z|x)dz, \qquad (2)$$

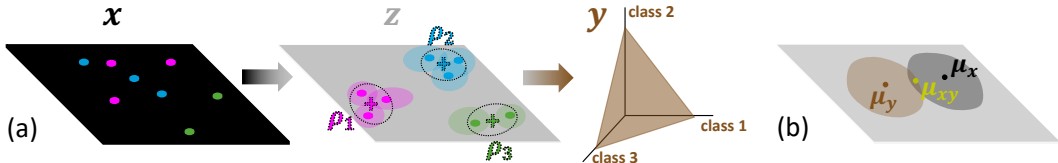

Figure 1: (a) Illustration of the stochastic prototype embedding. The model learns a mapping from input space, $\boldsymbol{x}$, to embedding space, $\boldsymbol{z}$, in which same-class instances are near and different-class instances are far. Embeddings are represented as Gaussian random variables. Prototypes, noted as + symbols in the embedding, are formed via a confidence-weighted average of the embeddings of instances known to belong to a class (support instances). Prototype uncertainty is depicted with the dotted ovals. Given the prototypes, a prediction of class $y$ is made for a query instance by marginalizing a softmax prediction over the embedding space. (b) Depiction of intersection sampler.

Informally, Figure 1a depicts the relationship between the input, latent, and class representations; for a formal depiction of SPE, see the chain graph in Appendix F. We train the SPE using the standard paradigm for supervised embedding methods, via a sequence of *episodes*, each with $m$ instances of $n$ classes. We split the $m \times n$ instances into $k \times n$ *support* examples, defining a set $S$, and $(m - k) \times n$ *query* examples. The support instances for each class $c$, $S_c \in S$, are used to determine the class prototype, $\boldsymbol{\rho}_c$, and the query instances are evaluated to predict class label (Equation 2).

## 3.1 Forming class prototypes

In the SPE, each class $y$ has an associated prototype in the embedding space, $\boldsymbol{\rho}_y$, and each instance $i$ of class $y$ projects to an embedding, $\boldsymbol{z}_i$, in the neighborhood of $\boldsymbol{\rho}_y$ such that:

$$p(\boldsymbol{z}_i|\boldsymbol{\rho}_y) = \mathcal{N}(\boldsymbol{\rho}_y, \sigma_\epsilon^2 \boldsymbol{I}). \tag{3}$$

With independent support instances and uniform class priors, Bayes rule obtains the posterior of prototype $\boldsymbol{\rho}_y$ for support set $S_y$:

$$p(\boldsymbol{\rho}_y|S_y) = \frac{\prod_{i \in S_y} p(\boldsymbol{z}_i|\boldsymbol{\rho}_y)}{\int_{\boldsymbol{\rho}} \prod_{i \in S_y} p(\boldsymbol{z}_i|\boldsymbol{\rho}) d\boldsymbol{\rho}}. \tag{4}$$

This normalized product of Gaussians is itself Gaussian:

$$p(\boldsymbol{\rho}_y|S_y) = \mathcal{N}(\boldsymbol{\mu}_y, \boldsymbol{\sigma}_y^2 \boldsymbol{I}) \text{ with } \boldsymbol{\sigma}_y^2 = \left(\sum_{i \in S_y} \hat{\boldsymbol{\sigma}}_{x_i}^{-2}\right)^{-1} \text{ and } \boldsymbol{\mu}_y = \boldsymbol{\sigma}_y^2 \circ \left(\sum_{i \in S_y} \hat{\boldsymbol{\sigma}}_{x_i}^{-2} \circ \boldsymbol{\mu}_{x_i}\right), \tag{5}$$

where $\hat{\boldsymbol{\sigma}}_{x_i}^2 = \boldsymbol{\sigma}_{x_i}^2 + \sigma_\epsilon^2$ and $\circ$ denotes the Hadamard product and $\{\boldsymbol{\mu}_x, \boldsymbol{\sigma}_x^2\}$ are the outputs of the embedding procedure for input $\boldsymbol{x}$. Essentially, the prototype is a confidence-weighted average of the support instances. This formulation has a clear advantage over the deterministic PN, which is premised on an unweighted average, because it de-emphasizes noisy support instances.

## 3.2 Prediction and approximate inference

Given a query embedding, $\boldsymbol{z}$, and uniform class priors, the class posterior is:

$$p(y|\boldsymbol{z}, S) \propto \mathcal{N}(\boldsymbol{z}; \boldsymbol{\mu}_y, \hat{\boldsymbol{\sigma}}_y^2 \boldsymbol{I}) \tag{6}$$

where $\hat{\boldsymbol{\sigma}}_y^2 = \boldsymbol{\sigma}_y^2 + \sigma_\epsilon^2$, and $\boldsymbol{\mu}_y$ and $\boldsymbol{\sigma}_y^2$ are the mean and diagonal covariance of $\boldsymbol{\rho}_y$ (Equation 5). Expressing this posterior in terms of the query $\boldsymbol{x}$ (i.e., combining Equations 1, 2, and 6):

$$p(y|\boldsymbol{x}, S) = \int_{\boldsymbol{z}} \mathcal{N}(\boldsymbol{z}; \boldsymbol{\mu}_x, \boldsymbol{\sigma}_x^2 \boldsymbol{I}) \frac{\mathcal{N}(\boldsymbol{z}; \boldsymbol{\mu}_y, \hat{\boldsymbol{\sigma}}_y^2 \boldsymbol{I})}{\sum_c \mathcal{N}(\boldsymbol{z}; \boldsymbol{\mu}_c, \hat{\boldsymbol{\sigma}}_c^2 \boldsymbol{I})} \, d\boldsymbol{z}. \tag{7}$$

The class distribution is equivalent to that produced by the deterministic PN as $\boldsymbol{\sigma}_x^2 \to \mathbf{0}$ when $\boldsymbol{\sigma}_y^2 = \boldsymbol{\sigma}_{y'}^2$ for all class pairs $(y, y')$. However, in the general case, the integral has no closed form solution; thus, we must sample to approximate $p(y|\boldsymbol{x}, S)$, both for training and evaluation. We employ two samplers, which we refer to as *naïve* and *intersection*.

### 3.2.1 NAÏVE SAMPLING

A direct approach to approximating the class posterior is to express Equation 2 as an expectation, $\mathbb{E}_{\boldsymbol{z} \sim p(\boldsymbol{z}|\boldsymbol{x})}\left[p(y|\boldsymbol{z}, S)\right]$, and to replace the expectation with the average over a set of samples. We utilize the reparameterization trick of Kingma and Welling (2014) to train the model. Although this Monte Carlo method is the simplest approach, it is sample-inefficient during training, and when the number of samples is reduced, model performance is impacted.

### 3.2.2 INTERSECTION SAMPLING

In Equation 7, the product of Gaussian densities in the numerator can be rewritten:

$$\mathcal{N}\left(\boldsymbol{z}; \boldsymbol{\mu}_x, \boldsymbol{\sigma}_x^2 \boldsymbol{I}\right) \mathcal{N}\left(\boldsymbol{z}; \boldsymbol{\mu}_y, \hat{\boldsymbol{\sigma}}_y^2 \boldsymbol{I}\right) = \mathcal{N}\left(\boldsymbol{z}; \boldsymbol{\mu}_{xy}, \boldsymbol{\sigma}_{xy}^2 \boldsymbol{I}\right) \mathcal{N}\left(\boldsymbol{\mu}_x; \boldsymbol{\mu}_y, (\boldsymbol{\sigma}_x^2 + \hat{\boldsymbol{\sigma}}_y^2)\boldsymbol{I}\right), \tag{8}$$

where $\boldsymbol{\sigma}_{xy}^2 = (\boldsymbol{\sigma}_x^{-2} + \hat{\boldsymbol{\sigma}}_y^{-2})^{-1}$ and $\boldsymbol{\mu}_{xy} = \boldsymbol{\sigma}_{xy}^2 \circ (\boldsymbol{\sigma}_x^{-2} \circ \boldsymbol{\mu}_x + \hat{\boldsymbol{\sigma}}_y^{-2} \circ \boldsymbol{\mu}_y)$. Substituting Equation 8 into Equation 7,

$$p(y|\boldsymbol{x}, S) = \mathcal{N}\left(\boldsymbol{\mu}_x; \boldsymbol{\mu}_y, (\boldsymbol{\sigma}_x^2 + \hat{\boldsymbol{\sigma}}_y^2)\boldsymbol{I}\right) \mathbb{E}_{\boldsymbol{z} \sim \mathcal{N}\left(\boldsymbol{\mu}_{xy}, \boldsymbol{\sigma}_{xy}^2 \boldsymbol{I}\right)} \left[\sum_c \mathcal{N}(\boldsymbol{z}; \boldsymbol{\mu}_c, \hat{\boldsymbol{\sigma}}_c^2 \boldsymbol{I})\right]^{-1}. \tag{9}$$

By approximating the expectation with samples from $\mathcal{N}\left(\boldsymbol{\mu}_{xy}, \boldsymbol{\sigma}_{xy}^2 \boldsymbol{I}\right)$, we obtain an elegant importance sampler that focuses on the intersection of the input distribution and a given class distribution, as illustrated in Figure 1b. During training with a cross-entropy loss, we need only sample for the known (target) class $y$. As we will demonstrate, this method is more robust and significantly more sample efficient than the naïve sampler, requiring only a *single* sample to train effectively.

## 4 EXPERIMENTAL RESULTS

We report on three sets of experiments. In Section 4.1, we demonstrate, using a synthetic data set, that SPE infers the generative structure of a domain, disentangles class-discriminating features, and provides meaningful estimates of label uncertainty and input noise. In Section 4.2, we show that SPE obtains state-of-the-art results on few-shot learning via a comparison to its deterministic sibling, PN, the previous state-of-the-art method. We evaluate on a standard data set used to compare methods in the few-shot learning literature, Omniglot (Lake et al., 2015). In Section 4.3, we show that SPE obtains state-of-the-art results on large-set classification via a comparison to the only other fully developed stochastic method for supervised embeddings, HIB (Oh et al., 2019). We evaluate on the only data set that Oh et al. (2019) used to explore HIB, a multi-digit variant of MNIST. For details regarding network architectures and hyperparameters, see Appendix A, and for simulation details, including the choice of initialization for $\sigma_\epsilon^2$, see Appendix B.

### 4.1 SYNTHETIC COLOR-ORIENTATION DATA SET

The data set consists of $64 \times 64$ pixel images of 'L' shapes, with four classes that are distinguished by orientation, color, or both (Figure 2a). Instances are sampled from a class-conditional isotropic Gaussian distribution in the generative space. (The isotropy of these qualitatively different dimensions comes from the fact that both can be mapped as directional quantities.) Because classes overlap on both color and orientation dimensions, elicited embeddings should indicate increased uncertainty near class boundaries. Full details of the synthetic data set can be found in Appendix A.2.

We trained a two-dimensional, intersection-sampling SPE on samples from this domain, using two instances per class to form prototypes. Classification accuracy of held-out samples is approximately $86\%$. Accounting for class overlap, a Bayes optimal classifier has an accuracy of approximately $87\%$. For visualization, Figure 2b presents a $5 \times 5$ array of examples with the class centroids in the corners and the other examples obtained by linear interpolation in the generative space. The resulting embeddings are presented in Figure 2c. Although the correspondence between Figures 2b and 2c seems trivial (mirror one set along the horizontal axis to obtain the other set), remember that the input space is $64 \times 64$ dimensional and the latent space is 2 dimensional. The network has captured the structure of the domain by disentangling the two factors of variation. Further, the embedding variance encodes label ambiguity; instances halfway between two classes on one dimension have maximal

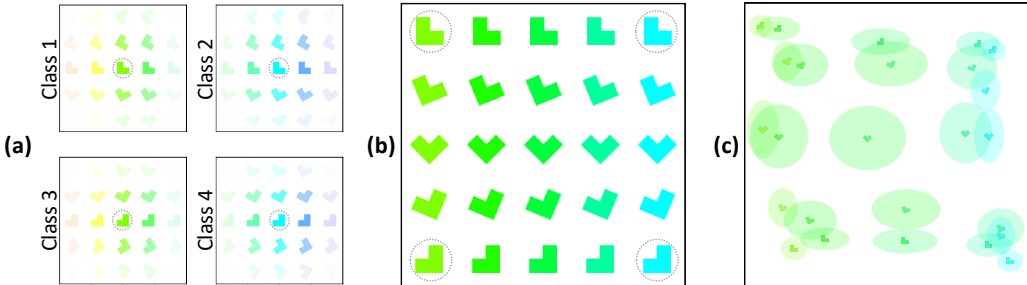

Figure 2: (a) Samples from the four classes in our synthetic data set. In each plot, class centroids are circled, along with samples spanning $\pm 2$ standard deviations in both orientation and color. A sample's transparency is set according to its class-conditional likelihood. Both dimensions can be coded as directional variables. The class centroids on each dimension are $90°$ apart with standard deviation of $30°$. (b) A set of examples, with the four class centroids located in the corners and other examples obtained by linear interpolation in the generative space. (c) The 2D stochastic prototype embedding for the examples in (b). The shape is plotted at the mean of $p(z|x)$, and the outlines of the ovals represent equiprobability contours at $0.4$ standard deviations.

variance along that dimension. Label ambiguity is one type of uncertainty. An equally important source of uncertainty comes from noisy or out-of-distribution (OOD) inputs. We examined OOD inputs generated in two different ways. In the left panel of Figure 3, we show the consequence of adding pixel hue noise to the four class centroids. Only one of these centroids is shown along the abscissa, but all four are used to make the graph, with many samples per noise level. The grey and black bars in the graph indicate variance on the horizontal and vertical dimensions of the embedding space, respectively. As pixel hue noise increases, uncertainty in color grows but uncertainty in orientation does not. In the right panel of Figure 3, we show the consequence of shortening the leg-length of the shape. Shortening the legs removes cues that can be used both for determining color and orientation. As a result, the uncertainty grows on both dimensions.

## 4.2 OMNIGLOT

The Omniglot data set contains images of labeled, handwritten characters from diverse alphabets. Omniglot is one of the standard data sets for comparing methods in the few-shot learning literature. The data set contains 1623 unique characters, each with 20 instances. Following Snell et al. (2017), each grayscale image is resized from $32 \times 32$ to $28 \times 28$, and we augment the original classes with all $90°$ rotations, resulting in 6492 total classes. We train PNs and SPEs episodically, where a training episode contains 60 randomly sampled classes and 5 query instances per class.

To compare the relative effectiveness of naïve and intersection samplers, we train the SPE on Omniglot varying both the sampler and the number of samples drawn per training query, denoted by $s$. We evaluate in a 1-*shot* 20-*class* setting, where shot refers to the number of support examples used to compute each prototype. Figure 4 shows test classification accuracy as the number of samples drawn per training trial ($s$) increases. As we previously stated, the intersection-sampling SPE is far more sample efficient, to the point that the intersection sampler with $s = 1$ outperforms the naïve sampler with $s = 81$. We have verified that the pattern in Figure 4 is consistent across simulations; consequently, we present only intersection-sampling SPE results in the remainder of the article, and

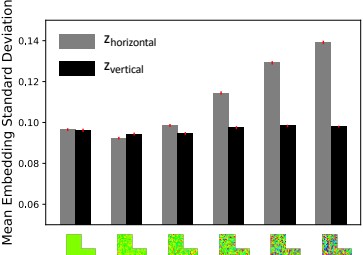

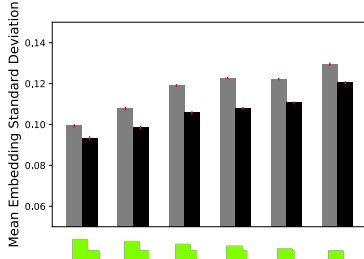

Figure 3: Synthetic data set: uncertainty on the two embedding dimensions as it becomes more difficult to discern the hue (left) and orientation (right).

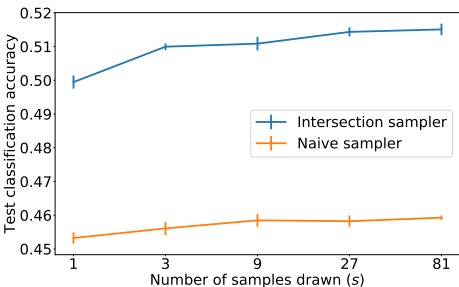

Figure 4: Test classification accuracy as a function of number of training samples per query instance for a naïve-sampling and intersection-sampling 2D SPE on a 1-shot, 20-class Omniglot task. Performance is a mean over 5 replications of running the model, showing $\pm 1$ standard error of the mean.

all SPEs are trained with a single sample ($s = 1$) per query. This choice causes the SPE to be on par with the PN in time and space requirements, even though using more samples may boost classification accuracy, as suggested by the trend in Figure 4.

Figure 5 is a visualization of a 2D embedding learned by the intersection-sampling SPE on Omniglot. All classes shown in the figure were held-out during training. Omniglot characters clearly vary along more than two dimensions, so a 2D SPE cannot learn a fully-disentangled representation as it did with the synthetic data set. However, we can still interpret the axes of the embedding. The horizontal axis appears to represent character complexity, with single-stroke characters on the left and many-stroke characters on the right. The vertical axis appears to encode the aspect ratio of the characters, with horizontally extended characters on the bottom and vertically extended characters on the top.

Figure 6a compares the PN and SPE with 2D embeddings on Omniglot test classes. Each bar is the mean accuracy across four conditions: 1-shot/5-class, 5-shot/5-class, 1-shot/20-class, and 5-shot/20-class. The first pair of bars perform the standard comparison in which the (1 or 5 instance) support set is used to obtain an embedding for each class, prototypes are formed, and query instances are classified. SPE is reliably better than the PN. Because the Omniglot data are carefully curated, the instances have little noise and therefore offer little opportunity to leverage SPE's assessment of uncertainty. Consequently, we corrupted instances by masking out rectangular regions of the input, as proposed by Oh et al. (2019). (See Appendix E for details.) The second and third sets of bars in Figure 6a correspond to the situations where the support and query instances are corrupted, respectively. SPE's advantage over PN increases significantly when the support instances are corrupted due to the fact that SPE's confidence-weighted prototypes (Equation 5) discount noisier support examples. Although the SPE is still superior when only the query is corrupted, the benefit is small. We also compared PN and SPE using a 64D embedding, but with high dimensional embeddings, both methods are near ceiling on this data set, resulting in comparable performance between the two methods. (See Appendix D for additional results, broken down by condition.)

To emphasize, SPE outperforms the PN, arguably the leading few-shot learning method, especially when inputs are corrupted, at essentially the same computational cost for training. And by providing an estimate of uncertainty associated with embedded instances, the SPE offers the possibility of detecting OOD samples and informing downstream systems that operate on the embedding.

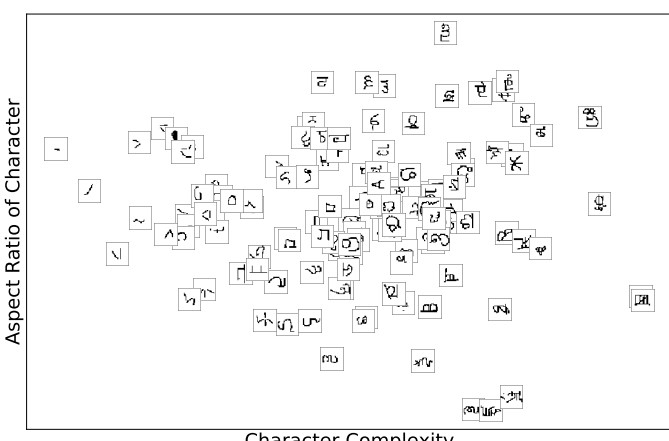

Figure 5: Two-dimensional embedding learned by the SPE on the Omniglot test set. Each square thumbnail image in the figure is a randomly-sampled instance from one of 125 randomly-sampled test classes and the location of the image represents the location of the class prototype. The images have a gray bounding box for visualization purposes only.

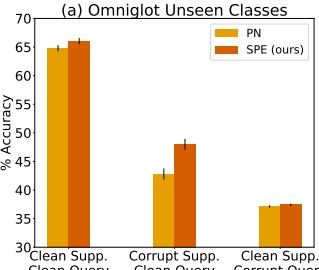 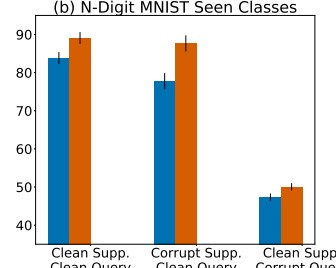 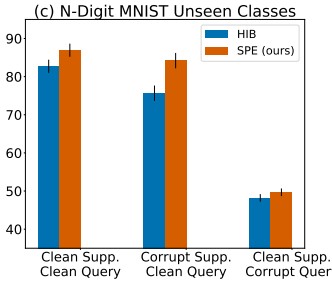

Figure 6: (a) Comparison of few-shot accuracy on Omniglot test classes for the PN (Snell et al., 2017) and our SPE. (b) Comparison of test accuracy on seen classes for $2$ and $3$-digit MNIST for HIB (Oh et al., 2019) and our SPE. (c) Same as (b) except for unseen classes. In (a)-(c), error bars reflect $\pm 1$ standard error of the mean, corrected to remove cross-condition variance (Masson and Loftus, 2003).

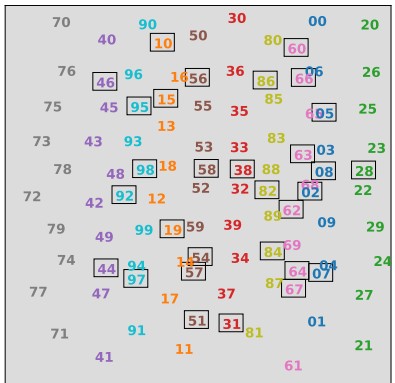 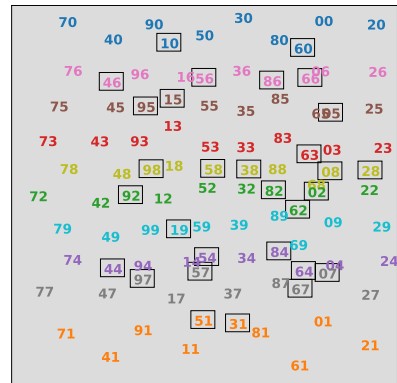

Figure 7: Two-dimensional embedding learned by the SPE on the 2-digit MNIST test set. A class is specified by a two-digit number. In both figures, the location of the class corresponds to the mean of the prototype in the test set using $140$ support instances. The digits surrounded by a black border are classes that were not seen during training. In the left and right figures, the prototypes are colored according to the first and second digit of the class, respectively.

## 4.3 N-DIGIT MNIST

The $N$-digit MNIST data set was proposed to evaluate HIB (Oh et al., 2019); it is formed by horizontal concatenation of $N$ MNIST digit images. The resulting images are $28 \times 28N$. To compare with HIB, we study $2$- and $3$-digit MNIST, and use a network architecture identical to that in Oh et al. (2019). Oh et al. (2019) split the data into a training set (with $70\%$ of the total classes), a *seen* test set, and an *unseen* test set. For 2-digit MNIST, the seen test set has the same 70 of 100 classes as the training set and the unseen test set has the remaining 30 classes. For 3-digit MNIST, the training set has 700 classes, the seen and unseen test sets each have a sample of 100 of the 700 seen or 300 unseen classes, respectively. We use the same train and test data splits as Oh et al. (2019), but we further divide the training split to include a validation set for early stopping.

Figure 7 shows two views of the 2D embedding learned by the SPE on the 2-digit MNIST test set. Each number is a class label; for example, 71, located in the lower left of the embedding, is the class in which the first of the two MNIST digits is a 7 and the second is a 1. The location of a label in the space corresponds to the mean of its prototype. In the left plot, each class is colored according to the first digit. The right plot is the same embedding, but each prototype is colored according to the second digit. The SPE learns an incredibly robust factorial representation in which the horizontal dimension represents the first digit of a class and the vertical dimension represents the second digit. A black bounding box indicates the unseen test classes, classes not presented during training. Impressively, the unseen test classes are embedded in exactly the positions where they belong, indicating that the SPE can discover relationships among classes that allow it to generalize to classes it has never seen during training. Furthermore, the embedding has captured inter-class similarity structure by placing visually similar digits close to one another. For example, on both the vertical and horizontal bands,

nines (teal) and fours (purple) are adjacent, and fives (brown) and threes (red) are adjacent. HIB discovers a clean decomposition along one dimension (Oh et al., 2019), but the second dimension is somewhat more entangled, suggesting that the SPE learns a more robust representation. Additionally, embeddings for the unseen class are not presented for HIB. The ability to sensibly embed novel classes is essential for any model that will be used for open-set recognition or few-shot learning. As we show in Appendix F, PNs do not obtain a clean compositional structure.

Figure 6b,c compare $N$-digit MNIST test accuracy on seen and unseen classes, respectively.[1] Each bar is the mean test accuracy across the Cartesian product of conditions specified by the number of MNIST digits in each image, $N \in \{2, 3\}$, and the dimensionality of the embedding, $D \in \{2, 3\}$. As in the Omniglot simulation, we varied whether support and query instances were clean or corrupted. The SPE outperforms HIB in all six comparisons. In the 24 individual conditions, SPE is worse on only 7. As in the Omniglot simulation, SPE shines best when support instances may be corrupted. (Appendix A.3 provides tabular results by condition, not only for HIB and SPE, but also their deterministic counterparts, contrastive loss and PN. Because the deterministic methods perform consistently worse than the stochastic methods, we omit the deterministic methods from the figure.)

Whereas SPE is a discriminative model with a specified classification procedure, Oh et al. (2019) had the freedom to design one. They use all available data—roughly 140 examples per class—and perform leave-one-out 5-nearest-neighbor classification. To be consistent with our episodic test procedure, the SPE uses only 50 support instances per class to form prototypes. It is particularly impressive that the SPE, based on a single stored prototype and approximately $1/3$ the labeled data, outperforms a nonparametric method that is able to model arbitrary distributions in latent space.

## 5 DISCUSSION AND CONCLUSIONS

We proposed the Stochastic Prototype Embedding (SPE) as a method for obtaining supervised embeddings which encode class-label uncertainty in their distributions. Such methods are useful for large- and open-set classification as well as few-shot learning, particularly for domains with ambiguous inputs or label noise. We compared SPE to the only fully-developed alternative method, the Hedged Instance Embedding (HIB), on *the complete battery of tasks* used to evaluate HIB. On these large-set classification tasks, SPE consistently outperforms HIB. Beyond its performance gains, SPE has no hand tuned parameters, whereas HIB has constant $\beta$ that determines characteristics of an information bottleneck (i.e., how much of the input entropy is retained in the embedding). Although one could simply set $\beta = 0$, doing so would encourage the net to perform like a softmax classifier and discard all information about inter-class similarity. Such similarities are essential in order to generalize to unseen classes (e.g., Figure 7).

SPE, which is a nondeterministic extension of the Prototypical Net (PN), matches or outperforms the PN on few-shot learning. Because the SPE extends the PN, it seems unlikely to fare worse; but because it can handle uncertainty in both the query and support set, it can fare better, particularly when the embedding space is low dimensional and the support instances may be corrupted. Extensions have been proposed to the PN compatible with ours (e.g., Allen et al., 2019); combining methods may potentially attain even stronger few-shot learning performance under uncertainty.

We proposed and evaluated an intersection sampler to train the SPE, which makes the SPE as time and space efficient for training as the deterministic PN, and more efficient for training than HIB, which relies on about 8 samples per item. (Our evaluation method for SPE presently involves drawing 200 samples from the naive sampler, though this conservative decision was arbitrary and not tuned.)

An unanticipated virtue of SPE is its ability to obtain interpretable, disentangled representations (Figures 2, 5, 7). Because uncertainty is encoded in a diagonal covariance matrix, any classification ambiguity maps to uncertainty in the value of individual features of the embedding. Thus, class-discriminating feature dimensions must align with the principle axes of the embedding space. In contrast to traditional unsupervised disentangling methods, which aim to discover the underlying generative factors of a domain, the SPE obtains a supervised analog in which the underlying class-discriminative factors are represented explicitly. This representation facilitates generalization to novel unseen classes and is therefore valuable for few-shot and lifelong learning paradigms.

---

[1]HIB results are from Oh et al. (2019). We thank the authors for providing us results on unseen classes, which were not included in their publication.

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

# A  NETWORK ARCHITECTURES AND HYPERPARAMETERS

## A.1  OMNIGLOT

For all Omniglot experiments, the network consisted of four convolutional blocks. The first three blocks had a convolutional layer with $64$ filters, a $3 \times 3$ kernel, zero-padding of length $1$, and a stride of $1$, followed by a batch normalization layer, ReLU activation, and $2 \times 2$ max-pooling. The fourth and final block had a convolutional layer with $2d$ filters, a $3 \times 3$ kernel, zero-padding of length $1$, and a stride of $1$, followed by $2 \times 2$ max-pooling, where $d$ represents the dimensionality of the embedding space. The flattened output of the network is a vector of length $2d$, where the first $d$ elements were considered the mean of the Gaussian distribution and the remaining $d$ elements were the diagonal covariance entries. The weights were initialized using He initialization and the biases with the following uniform distribution: $\mathcal{U}(-\frac{1}{\sqrt{\text{fan in}}}, \frac{1}{\sqrt{\text{fan in}}})$.

All Omniglot models were trained with an initial learning rate of $0.001$ which was cut in half every $50$ epochs. The models were stopped early using a patience parameter when performance on the validation set no longer increased.

## A.2  SYNTHETIC DATA

The images in the synthetic data set are $64 \times 64$ pixels in size. For orientation, we chose class centers at $90°$ and $180°$, with a standard deviation of $30°$. For color, we manipulated the hue and kept value and saturation constant. Like orientation, hue is a circular quantity. If hue ranges from $0$ to $360$ degrees, we chose color class centers and standard deviation in the same way as orientation. Additionally, we add noise to a minority (15%) of the images used to train the model. For these, we add Gaussian noise to the hue of each pixel inside the shape. The standard deviation of the hue noise was chosen uniformly between $18°$ and $54°$. We also added noise to the leg lengths of the L shapes. The leg length was chosen uniformly between 10% and 98% of its original length. See Figure 3 for some examples.

The network followed an architecture similar to the one we used for Omniglot, except that we added two additional blocks of convolution, batch normalization, ReLU, and max-pooling because the images are larger. We used $2$ instances per class to form prototypes and $8$ samples per query instance during training. We used a learning rate of $0.0001$ and the models were stopped early using a patience parameter when performance on the validation set no longer increased.

## A.3  N-DIGIT MNIST

For all $N$-digit MNIST experiments, we constructed an architecture which we believe to be identical to that used for HIB MNIST experiments, based on code provided by the authors (Oh et al., 2019). The network consisted of two convolutional blocks followed by two fully-connected layers. The convolutional blocks each contained a convolutional layer, followed by an ReLU activation, and $2 \times 2$ max-pooling. The first convolutional layer had $6$ filters, a $5 \times 5$ kernel, zero-padding of length $2$, and a stride of $1$. The second convolutional layer was identical to the first, but had $16$ filters instead of $6$. The output of the second convolutional block was flattened, passed through a fully-connected layer with $120$ units, an ReLU activation, and a final fully-connected layer with $2d$ units, where $d$ represents the dimensionality of the embedding space. Like the Omniglot architectures, the first $d$ entries in the output vector are treated as the mean and the remaining $d$ elements as the diagonal covariance entries. The weights were initialized using a Xavier uniform initialization and biases were initialized to zero.

The PN and SPE are trained episodically with all performance results in the main article measured as the mean over $1000$ random test episodes. All $N$-digit MNIST models were trained with an initial learning rate of $0.001$ which was cut in half every $50$ epochs. The models were stopped early using a patience parameter when performance on the validation set no longer increased. For 2-digit MNIST, each episode in training, validation, and seen-class testing contained all $70$ classes and $50$ support instances per class. For testing of unseen classes, each episode contained all $30$ classes. For 3-digit MNIST, each episode contained $100$ classes and either $20$ support instances per class for training/validation or $50$ support instances per class for seen- and unseen-class testing.

## B    SIMULATION DETAILS

For all SPE models,

$$\sigma_\epsilon^2 = \mathrm{softplus}\left(\gamma\right),$$

where $\gamma$ is a trainable parameter. We initialize $\gamma$ using the following prescription:

$$\gamma = |S|\gamma_0^{2/d},$$

where $|S|$ is the number of support examples per episode during training and $d$ is the dimensionality of the embedding. We chose this prescription for two reasons: (1) as the number of support examples increases, the variance of the prototype distribution approaches zero, so scaling linearly by $|S|$ tends to provide a stronger training signal early on, and (2) the amount of noise in the projection of an embedding should scale with the dimensionality of the embedding space as to maintain unit-volume. All models used $\gamma_0 = 0.01$.

The variance of each dimension $i$, $\sigma_{x_i}^2$, is guaranteed to be non-negative by using a softplus transfer function.

Whether trained with the naïve or intersection sampler, we evaluate model performance using the naïve sampler with 200 samples. This approach ensures that we are comparing the quality of models based only on the method by which they were trained.

## C    SPE VARIANTS

We assumed only diagonal covariance matrices in this work. Switching to a full covariance matrix would require matrix inversion, which is ordinarily infeasible, but because one purpose of deep embeddings is visualization, there may be interesting cases involving 2D embeddings where the cost of inversion is trivial. However, using a diagonal covariance matrix causes class-discriminating features to be aligned with the axes of the latent space, as we argued in the main article, and this alignment is a virtue for interpretation.

# D  Tabular Results

## D.1  Omniglot

Table 1: Test classification accuracy (%) on Omniglot with both a 2D and 64D embedding for clean-support/clean-query, corrupt-support/clean-query, and clean-support/corrupt-query. PN is our implementation of Prototypical Networks (Snell et al., 2017). SPE is our model. SPE is trained with intersection sampling (1 sample per trial). Reported accuracy for each experimental configuration is the mean over 1000 random test episodes.

**2D Clean Support, Clean Query**

|  | 1-shot, 5-class | 5-shot, 5-class | 1-shot, 20-class | 5-shot, 20-class | Mean |
|---|---|---|---|---|---|
| PN | 75.7 | 82.6 | 45.0 | 55.9 | 64.8 |
| SPE | 76.9 | 82.3 | 49.7 | 55.3 | 66.1 |

**2D Corrupt Support, Clean Query**

|  | 1-shot, 5-class | 5-shot, 5-class | 1-shot, 20-class | 5-shot, 20-class | Mean |
|---|---|---|---|---|---|
| PN | 50.0 | 65.9 | 23.6 | 31.7 | 42.8 |
| SPE | 50.7 | 73.9 | 25.6 | 41.6 | 48.0 |

**2D Clean Support, Corrupt Query**

|  | 1-shot, 5-class | 5-shot, 5-class | 1-shot, 20-class | 5-shot, 20-class | Mean |
|---|---|---|---|---|---|
| PN | 48.9 | 52.3 | 21.7 | 25.6 | 37.1 |
| SPE | 47.8 | 52.3 | 22.8 | 26.8 | 37.4 |

**64D Clean Support, Clean Query**

|  | 1-shot, 5-class | 5-shot, 5-class | 1-shot, 20-class | 5-shot, 20-class | Mean |
|---|---|---|---|---|---|
| PN | 98.5 | 99.6 | 94.9 | 98.6 | 97.9 |
| SPE | 98.5 | 99.5 | 94.9 | 98.6 | 97.9 |

**64D Corrupt Support, Clean Query**

|  | 1-shot, 5-class | 5-shot, 5-class | 1-shot, 20-class | 5-shot, 20-class | Mean |
|---|---|---|---|---|---|
| PN | 85.0 | 98.7 | 68.5 | 95.6 | 87.0 |
| SPE | 85.7 | 98.8 | 69.3 | 95.7 | 87.4 |

**64D Clean Support, Corrupt Query**

|  | 1-shot, 5-class | 5-shot, 5-class | 1-shot, 20-class | 5-shot, 20-class | Mean |
|---|---|---|---|---|---|
| PN | 80.9 | 84.9 | 66.7 | 74.7 | 76.8 |
| SPE | 80.3 | 84.8 | 66.3 | 74.7 | 76.5 |

## D.2    N-Digit MNIST

Table 2: Test classification accuracy (%) on 2- and 3-digit MNIST for clean-support/clean-query, corrupt-support/clean-query, and clean-support/corrupt-query. $N$: number of digits in each image; $D$: dimensionality of the embedding. Contrastive and HIB results from Oh et al. (2019). PN is our implementation of Prototypical Networks (Snell et al., 2017). SPE is our model. SPE is trained with intersection sampling (1 sample per trial). Reported accuracy for PN and SPE for each experimental configuration is the mean over 1000 random test episodes.

CLEAN SUPPORT, CLEAN QUERY

|  | SEEN TEST CLASSES | | | | | UNSEEN TEST CLASSES | | | | |
| --- | --- | --- | --- | --- | --- | --- | --- | --- | --- | --- |
|  | N=2 | | N=3 | | | N=2 | | N=3 | | |
|  | D=2 | D=3 | D=2 | D=3 | MEAN | D=2 | D=3 | D=2 | D=3 | MEAN |
| CONTRASTIVE | 88.2 | 95.0 | 65.8 | 87.3 | 84.1 | 85.5 | 84.8 | 59.0 | 85.5 | 78.7 |
| HIB | 87.9 | 95.2 | 65.0 | 87.3 | 83.9 | 87.3 | 91.0 | 64.4 | 88.2 | 82.7 |
| PN | 91.1 | 95.0 | 65.8 | 90.6 | 85.6 | 82.0 | 89.5 | 64.3 | 89.1 | 81.2 |
| SPE | 93.0 | 94.2 | 80.2 | 89.0 | 89.1 | 90.0 | 89.3 | 80.2 | 88.2 | 86.9 |

CORRUPT SUPPORT, CLEAN QUERY

|  | SEEN TEST CLASSES | | | | | UNSEEN TEST CLASSES | | | | |
| --- | --- | --- | --- | --- | --- | --- | --- | --- | --- | --- |
|  | N=2 | | N=3 | | | N=2 | | N=3 | | |
|  | D=2 | D=3 | D=2 | D=3 | MEAN | D=2 | D=3 | D=2 | D=3 | MEAN |
| CONTRASTIVE | 76.2 | 92.2 | 49.5 | 77.6 | 73.9 | 76.5 | 73.3 | 42.6 | 73.2 | 66.4 |
| HIB | 81.6 | 94.3 | 54.0 | 81.2 | 77.8 | 80.8 | 86.7 | 53.9 | 81.2 | 75.7 |
| PN | 72.7 | 93.3 | 44.6 | 82.7 | 73.3 | 70.9 | 86.3 | 42.9 | 79.6 | 69.9 |
| SPE | 92.4 | 93.8 | 76.7 | 87.8 | 87.7 | 88.8 | 86.3 | 75.4 | 86.3 | 84.2 |

CLEAN SUPPORT, CORRUPT QUERY

|  | SEEN TEST CLASSES | | | | | UNSEEN TEST CLASSES | | | | |
| --- | --- | --- | --- | --- | --- | --- | --- | --- | --- | --- |
|  | N=2 | | N=3 | | | N=2 | | N=3 | | |
|  | D=2 | D=3 | D=2 | D=3 | MEAN | D=2 | D=3 | D=2 | D=3 | MEAN |
| CONTRASTIVE | 43.5 | 51.6 | 29.3 | 44.7 | 42.3 | 46.3 | 44.8 | 26.2 | 42.0 | 39.8 |
| HIB | 49.9 | 57.8 | 31.8 | 49.9 | 47.4 | 53.5 | 57.0 | 32.1 | 50.2 | 48.2 |
| PN | 53.1 | 61.1 | 33.8 | 56.4 | 51.1 | 51.1 | 57.9 | 33.0 | 54.8 | 49.2 |
| SPE | 53.7 | 58.2 | 40.2 | 48.1 | 50.1 | 56.3 | 56.5 | 39.3 | 46.6 | 49.7 |

# E    CORRUPTION PROCEDURE

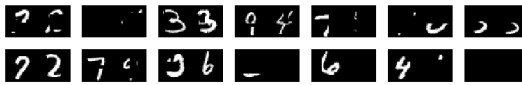

Figure 8: Examples of occluded 2-digit sequences. Occlusion is based on random rectangles that black out portions of each digit.

The algorithm for applying corruption was identical to the scheme used in Oh et al. (2019). A random rectangular-sized occlusion of black pixels was determined by first sampling a patch width, $L_x$, and patch height, $L_y$, from a uniform distribution, $L_x, L_y \sim \mathcal{U}(0, 28)$, and then sampling the top-left corner coordinates, $TL_x \sim \mathcal{U}(0, 28 - L_x)$, $TL_y \sim \mathcal{U}(0, 28 - L_y)$. This resulted in an occlusion of area $L_x \times L_y$. Note that if $L_x = 0$ or $L_y = 0$, the image was left unoccluded. Figure 8 shows examples of occluded 2-digit images.

For Omniglot, we only trained/validated on corrupted imagery if the test set contained a corrupted support or corrupted query set. When testing on clean support and clean query, the training and validation sets were left unoccluded. When testing on corrupted imagery, the training and validation sets corrupted each character independently with a probability of 0.2.

The training and validation sets for $N$-digit MNIST corrupted each digit of each image independently with a probability of 0.2, regardless of test imagery. This matched Oh et al. (2019).

During testing on both data sets, we considered both clean and corrupt support sets, as well as clean and corrupt query sets. A clean set was one in which all digits/characters were unoccluded. A corrupt set occluded each digit/character in each image according to the procedure described above.

## F VISUALIZATIONS OF EMBEDDINGS DISCOVERED BY PROTOTYPICAL NETWORKS

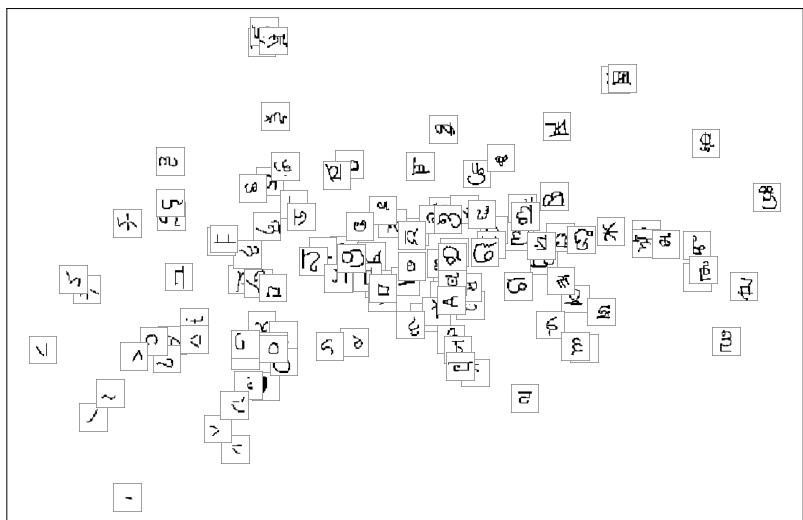

Figure 9: Two-dimensional embedding learned by the PN on the Omniglot test set. Each square thumbnail image in the figure is a randomly-sampled instance from one of 125 randomly-sampled test classes and the location of the image represents the location of the class prototype. The images have a gray bounding box for visualization purposes only.

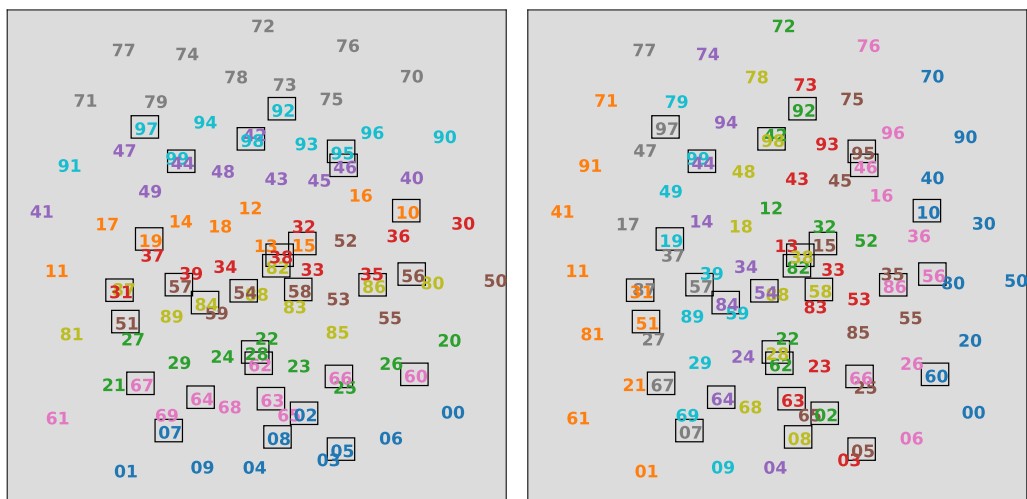

Figure 10: Two-dimensional embedding learned by the PN on the 2-digit MNIST test set. A class is specified by a two-digit number. In both figures, the location of the class corresponds to the mean of the prototype in the test set using 140 support instances. The digits surrounded by a black border are classes that were not seen during training. In the left and right figures, the prototypes are colored according to the first and second digit of the class, respectively.

## G GRAPHICAL MODEL UNDERLYING SPE

Figure 11 presents SPE as a chain graph with both directed and undirected edges. The left column of the Figure depicts the relationship between the support set and the prototypes; the plates are instantiated for each class, $c$, and each support example within a class, $s$. The right column depicts the relationship between a query instance and its class label.

The notation in the Figure is consistent with the notation in the main article, except that we have split the latent representation into a bottom up embedding, $z$, and a top-down embedding $z^\epsilon$. The two representations are constrained by the corruption relationship mentioned in the main article:

$$z^\epsilon \sim \mathcal{N}(z, \sigma_\epsilon^2 I) \text{ or equivalently, } z \sim \mathcal{N}(z^\epsilon, \sigma_\epsilon^2 I).$$

The $z^\epsilon$ nodes are included for clarity, but they are deterministic functions of their parents:

$$z_{sc}^\epsilon = \rho_c \text{ and } z_q^\epsilon = \rho_y.$$

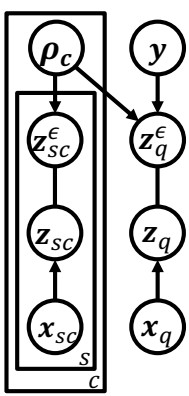

Figure 11: Chain graph representing SPE

The model proposed by Fort (2017) shares with SPE the notion that prototypes are a confidence-weighted average of support instances (Equation 5), and the embedding procedure produces a scaling matrix for computing a Mahalanobis distance, similar to the Gaussian covariance matrix of SPE. However, Fort (2017) does not treat the embedding as stochastic, for to do so, it would need to marginalize over the uncertainty in the embedding to predict a class label. This marginalization is the core of a probabilistic model and is the critical component of SPE. Fort (2017) also obtain best results with a spherical scaling matrix for the Mahalanobis distance, whereas a critical property of SPE is that the uncertainty varies on each dimension of the latent space; our disentangling and uncertainty results all hinge on using a more flexible diagonal covariance matrix. And Fort (2017) omits the $\sigma_\epsilon$ noise term which turns out to be critical both to obtain a well-formed probabilistic model (Figure 11) and for the model to work in practice.

