# OpenReview forum: "Stochastic Prototype Embeddings"
_ICLR.cc/2020/Conference — Reject_

### Official Review · AnonReviewer2 · 2019-10-22
**Official Blind Review #2**

**Rating:** 3

**Review:**

The paper proposes stochastic prototype embeddings (SPE) for few-shot learning. The method is an extension of Prototypical Networks (PN, [1]) with Gaussian embeddings. The idea is to take representation uncertainty into account when classifying objects which makes the model more robust to input and label noise. The authors propose an efficient sampling algorithm to train SPE which outperforms naive Monte Carlo sampling. They conduct a range of experiments on few-shot learning tasks on a synthetic dataset, Omniglot and N-digit MNIST and compare to Prototypical Networks and previous stochastic embedding state-of-the-art method HIB [3]; SPE was shown to outperform prior work in most settings. The authors also plot interpretable disentangled representations learned in 2D embedding space.

However, in my opinion there are a few weaknesses in the paper in its current state: (1) the clarity of sections 3.1-3.2 could be significantly improved as currently the proposed probabilistic model framework is confusing and not well-defined; (2) the experimental results are provided only for embedding spaces of dimensionality 2-3 which significantly hinders performance of Prototypical Networks compared to having a much higher dimensional embedding space, so it would help to see comparisons of SPE and PN using high-dimensional embeddings; (3) the central idea and proposed model seem to be very close to those of [2] which is mentioned in the related work, so, please, list differences with this prior work in the updated version.
For these reasons, I recommend a weak reject for the paper. I explain these issues in more detail below.

(1) SPE model assumes that each embedding z is Gaussian-distributed p(z|x) = N(z; mu_x, sigma^2_x I) where parameters of the distribution mu_x and sigma_x are outputted by an embedding neural network. In equation (3) the author define each class prototype as rho_y = z_i + eps where z_i are instances of class y and eps is noise ~ N(0, sigma^2_eps I), so p(rho_y | z_i) = N(rho_y | z_i, sigma^2_eps I) for every z_i from class y. This definition is confusing to me since rho_y is redefined for each z_i, so it is not clear what the generative process for rho_y is. In the Introduction section it is mentioned “...each class instance is assumed to be a Gaussian perturbation of the prototype” which suggests z_i = rho_y + eps for every z_i, so graphical model would be rho_y -> z_i instead of z_i -> rho_y implied in section 3.1. Please, clarify in the rebuttal which the graphical model is implied in SPE. Is it x_i -> z_i -> rho_y? How is then the distribution of rho_y defined given all z_i from class y?
Further, in the equation (4) the likelihood of rho_y given all x_i from class y is proportional to product of p(rho_y | x_i) so the expression was factorized over the condition which is the reflection of the authors’ assumption of consistency of the prototype. p(rho_y | x_1, … x_n) could be also written as proportional to the product of p(x_i | rho) times p(rho) using Bayes rule. It would help if authors could explain the differences between the two possible expressions for p(rho_y | x_1 … x_n) and why the former is chosen.
Also, in equation (6) the right-hand-side is p(z | y, S) = p(z | rho_y, S) = N(z | mu_y, \hat{sigma}^2_y I): why is the variance is \hat{sigma}^2_y (so extra sigma^2_eps added to sigma^2_y) instead of just sigma^2_y defined in equation (5)? The noise term with sigma^2_eps seems to be already included in sigma^2_y.

(2) In all presented experiments, the dimensionality of the embedding is 2-3 instead of high dimensional embedding spaces in the original Prototypical Network paper which causes a significant drop in performance: e.g., on Omniglot 1-shot 5-class classification, the accuracy dropped from 98.8% to 75.7%. SPE outperformed PN, but an important aspect of the PN model, embedding space dimensionality, was changed from the original in a way that hindered the performance very much, so SPE’s advantage is unclear. It is mentioned in section 4.2 that “We also compared PN and SPE using a 64D embedding, but with high dimensional embeddings, both methods are near ceiling on this data set, resulting in comparable performance between the two methods. (See Appendix D for additional results, broken down by condition.)” but the results on 64D embeddings are not added and discussed in the appendix. If the performance on high dimensional embeddings in the standard setting is similar for SPE and PC, it would help to see the 64D embedding experiment with corrupted support and query data, because if SPE outperformed PN on the corrupted data setting, it would be a fair advantage. It is interesting to see what the 2D embedding space looks like in the visualizations (Figures 2, 5, 7), and SPE is learning interpretable disentangles representations. However, for better classification performance it makes sense to use higher dimensional embedding space to allow learning more expressive representations.
For Figures 2c, 5 and 7, it would be helpful to see how the embedding space of PN looks like and compare it visually to the one of SPE, e.g. whether PN has disentangled representations. It seems like we could get a similar embedding space to, for example, Figure 2c and the label uncertainty would come from the roughly equal distance of object representation to different class prototypes. Please, add respective embedding visualizations for PN and explain what the advantages of SPE learned representations over PN’s representations are.

(3) The prior work of [2] referenced in the paper seems to have a very similar method (judging from Figure 2 and equations 5-6 which compute parameters of the prototype distribution, [2]). Please, list the differences with this prior work and provide experimental comparison if possible.


Other questions:
1. In Figure 2c, please plot and highlight the learned distribution of the class prototypes p(rho | S).
2. Section 4.1: did each object in the training set have only one label or multiple labels (since classes are overlapping)?
3. Experiments: please add HIB method to synthetic data and Omniglot experiments if possible. It would also be interesting to compare to PN on miniImageNet (section 3.2 of [1]). Another interesting experiment which could show advantage of SPE could be out-of-distribution data detection (e.g., comparing likelihoods p(z | y) for in-distribution representations z and out-of-distribution z).


[1] Snell, Jake, Kevin Swersky, and Richard Zemel. "Prototypical networks for few-shot learning." Advances in Neural Information Processing Systems. 2017.
[2] Fort, Stanislav. "Gaussian prototypical networks for few-shot learning on omniglot." arXiv preprint arXiv:1708.02735 (2017).
[3] Oh, Seong Joon, et al. "Modeling uncertainty with hedged instance embedding." arXiv preprint arXiv:1810.00319 (2018).

**Experience Assessment:**

I have read many papers in this area.

**Review Assessment: Checking Correctness Of Derivations And Theory:**

I carefully checked the derivations and theory.

**Review Assessment: Checking Correctness Of Experiments:**

I carefully checked the experiments.

**Review Assessment: Thoroughness In Paper Reading:**

I read the paper thoroughly.

---

> ### Author Response · Authors · 2019-11-13
> **Response to Review #2**
>
> We thank the reviewer for their feedback.
>
> We have completely rewritten our description of the model in a manner that we hope will alleviate the reviewer’s concerns (Sections 3.1, 3.2). We felt the same frustration as the reviewer but originally believed that we needed to express SPE as a recognition model ($x \rightarrow z \rightarrow y$), whereas the elegant (Bayesian) formulation of prototype formation and classification follows from a generative model ($x \leftarrow z \leftarrow y$). As we explain in a new section in the appendix which shows the underlying graphical model, we are able to express the SPE as a chain graph, consisting of both directed and undirected links ($x \rightarrow z -  z^{\epsilon} \leftarrow y$).
>
> You asked why Equation 6 includes an ‘extra’ $\sigma^2_{\epsilon}$. The reason is that for any prototype, $\rho_y \sim \mathcal{N}(\mu_y, \sigma^2_y)$, the embedding z will have Gaussian corruption (Equation 3). The noise influences each embedding that forms the prototype as well as each embedding to be classified. This point is clearer by viewing the graphical model we’ve included in Appendix G.
>
> We have added to Appendix D a comparison of PN and SPN on few-shot learning for 64D embeddings. Apologies for the omission. We have also added a comparison for the 64D embedding with noise corruption. Our reason for focusing on low-dimensional embeddings is that the HIB paper found little performance benefit of a stochastic representation with high dimensional embedding spaces; they focused on 2- and 3-dimensional embeddings. Presumably with a more complex domain and more complex noise sources, one might expect to see benefits of stochastic representations even in higher dimensional spaces. However, neither we nor the HIB paper have investigated this assumption thoroughly.
>
> We have added visualizations of the PN embeddings on Omniglot and 2-digit MNIST (Appendix F). Thank you for the suggestion. Because PNs use a Euclidean metric, there is necessarily no alignment of the axes with class-discriminating features. But, even allowing for rotations of the space, our visualization reveals far less organization of the classes for PN embeddings.
>
> We agree with the reviewer that the Fort model is a step along the way from PN to SPE, but it is a small step because it lacks probabilistic semantics. We frame SPE throughout in well-formed probabilistic terms (both in our submitted paper and in the improved formulation in our revision, prompted by your feedback).  Fort essentially proposes using a Mahalanobis distance instead of a Euclidean distance to assess similarity. Fort does not treat the embedding as stochastic, for to do so, it would need to marginalize over the uncertainty in the embedding to predict a class label. This marginalization is the core of a probabilistic model and is the critical component of SPE. Fort also finds that the version of his model that works best uses a spherical scaling matrix for the Mahalanobis distance, whereas a critical property of SPE is that the uncertainty varies on each dimension of the latent space; our disentangling and uncertainty results all hinge on using a more flexible diagonal covariance matrix. And Fort omits the $\sigma_\epsilon$ noise term which turns out to be critical both to obtain a well-formed probabilistic model (see Appendix G) and for the model to work in practice. Fort does report improvements over PNs, but the comparisons are not apples-to-apples as Fort endows his model with a larger backbone architecture and a larger embedding dimension--significant factors in few-shot learning performance.  Nonetheless, we do agree that there are some similarities and we have revised our paper to give better acknowledgement to Fort (Section 2).
>
> In Section 4.1, examples are drawn IID from the class-conditional distribution and assigned a deterministic label for that class. Thus, each example has exactly one class label but it is theoretically possible that in the region of class overlap, the same point could be sampled to represent 2 distinct classes.
>
> We cannot easily run HIB on additional experiments, as much as we would like to. The HIB code has not been open-sourced, and we are not confident of some implementation details.  As it was, we had in-depth interactions with the authors of HIB just to ensure we could reproduce their experimental conditions; these interactions resulted in the HIB authors submitting a revision of their paper.

---

### Official Review · AnonReviewer1 · 2019-10-23
**Official Blind Review #1**

**Rating:** 3

**Review:**

Authors propose Stochastic Prototype Embeddings, as a probabilistic extension of Prototype Networks (snell et al, 2017) for few-shot classification. Authors claim their method leads to better few-shot results in the case of corrupted data, and support their claims with a fair number of results over N-Mnist, with 'Hedge Instance Embeddings.


I think overall is a good paper as it contains interesting new ideas and represents thorough work, including their empirical validation. However, I believe it is not strong enough for me to recommend acceptance at this point. I hope the authors will address my concerns

1)The derivation of their probabilistic method is not satisfactorily explained from a statistical perspective. Why do they appeal to a product distribution? the kind of formulae the authors obtain (e.g. equation 4) resembles the usual ones for posterior distributions over gaussians. Instead of talking about a 'product distribution' it would be much better if the authors appealed to statistical principles so that their choices reveal themselves as sensible or natural. Additionally, their "intersection sampling" seems as rebranding of usual importance sampling. I hope the authors will comment more on the originality of their approach, and if not original, downplay the contribution of the sampler.

2)Although results are solid, but when going through the results section I got the impressions the authors were not clear about what were their ultimate intentions, what they wanted to prove. Key results are presented in Figure 6 but they led me with the following questions that I hope the authors will be able to better respond. Is the main preoccupation about few(or zero)-shot learning? then, is HIB the proper baseline in figure 6 and 7? My concern comes from the fact that (to my understanding) HIB isn't stated in the context of few shot learning so it seems authors are defeating a straw man.  uthors may compare with a more naive baseline for uncertainty modeling. Alternatively, authors may compare with HIB in non-few shot regimes. In any case, I hope the authors will make this point clear.

**Experience Assessment:**

I do not know much about this area.

**Review Assessment: Checking Correctness Of Derivations And Theory:**

I assessed the sensibility of the derivations and theory.

**Review Assessment: Checking Correctness Of Experiments:**

I assessed the sensibility of the experiments.

**Review Assessment: Thoroughness In Paper Reading:**

I read the paper at least twice and used my best judgement in assessing the paper.

---

> ### Author Response · Authors · 2019-11-13
> **Response to Review #1**
>
> We thank the reviewer for their feedback.
>
> We have completely rewritten our description of the model in a manner that we hope the reviewer will find satisfactory from a statistical perspective. We agree with the reviewer’s concerns, but we originally believed that we needed to express SPE as a recognition model ($x \rightarrow z \rightarrow y$), whereas the elegant (Bayesian) formulation of prototype formation and classification follows from a generative model ($x \leftarrow z \leftarrow y$). As we explain in a new section in the appendix which shows the underlying graphical model, we are able to express the SPE as a chain graph, consisting of both directed and undirected links ($x \rightarrow z -  z^{\epsilon} \leftarrow y$). Note that our new description does not change the model; it gives the model a more elegant framing.
>
> Our intersection sampler is indeed an importance sampler, and we have updated the paper to indicate such (Section 3.2.2). However, we feel the sampler does deserve a distinct name given that the sampling distribution and reweighting term follow from the fact that our classes and embeddings are Gaussian, and importance sampling actually results in a simplified formulation (because the weighting term cancels a term in the integral). We thank the reviewer for pointing out this oversight.
>
> The reviewer feels that we have not been clear in stating our aims, and we have modified the paper to be more explicit (last para of Section 1, first para of Section 5). To elaborate here, we develop a model (SPE) that extends an existing model (PN) to discover supervised embeddings that are random variables. Our primary contribution is to demonstrate that allowing uncertainty in the embeddings improves model performance in a supervised-learning task when inputs are ambiguous or class labels are noisy. To this end, we compare to the only other fully-formed method for stochastic embeddings, HIB, on the complete set of tasks that were used to evaluate HIB. Our secondary contribution is to establish that SPE is a viable alternative to PNs. The appropriate domain for comparison of SPE and PNs is the domain in which PNs were developed: few-shot learning. Our findings suggest that SPE is strictly equal or superior to PNs in performance, involves little additional computational cost, and attains more interpretable (disentangled) representations. All three of the models (SPE, PN, HIB) are supervised embedding methods, which are—as we state in the paper’s introduction—suited for few-shot learning, open-set classification, and large-set classification tasks.

---

### Official Review · AnonReviewer3 · 2019-10-29
**Official Blind Review #3**

**Rating:** 3

**Review:**

By extending prototypical networks, this paper proposes a probabilistic model, i.e., stochastic prototype embedding, that treats embeddings as random variables. The model is very straightforward and easy to understand. The authors make a few assumptions to simplify the problem. For example, the distance between every instance $z_i$ of class $y$ and the class embedding $\rho_y$ follows a Gaussian distribution, and the a softmax prediction for a query embedding also follows a Gaussian distribution. Combining these, the authors give the class posterior for the query. Since the class posterior involves an integral, the paper employs a naive samplying and an intersection sampling. The intersection sampling seems a bit interesting in the sense that the sampler focuses on the intersection of the input distribution and the class distribution and it is more sample-efficient.

In general, I think this probabilistic approach is very natural and simple, which is definitely one of the advantages. Compared to the deterministic approach, i.e., prototypical networks, one biggest advantage I can think of is that it will be more robust while dealing with noisy training set (many outliers exist). However, such probabilistic formulation does not seem to be new in terms of visual recognition, although it may be new in few-shot learning.

I also have a few concerns. One of my biggest concerns is the experiment. The experiments conducted in the paper are toy-ish. I can see that the proposed method indeed shows some gains over the prototypical networks on some toy tasks, but one can not really tell whether this approach really works for more realistic settings. For few-shot learning, I believe the authors should try to run experiments on Mini-ImageNet or CUB. A good experimental example to follow is [A Closer Look at Few-shot Classification, ICLR 2019]. It should be easy to run your model in their settings, since they have oper-sourced code.

**Experience Assessment:**

I have published one or two papers in this area.

**Review Assessment: Checking Correctness Of Derivations And Theory:**

I assessed the sensibility of the derivations and theory.

**Review Assessment: Checking Correctness Of Experiments:**

I assessed the sensibility of the experiments.

**Review Assessment: Thoroughness In Paper Reading:**

I read the paper at least twice and used my best judgement in assessing the paper.

---

> ### Author Response · Authors · 2019-11-13
> **Response to Review #3**
>
> We thank the reviewer for their feedback.
>
> We did a poor job of emphasizing our primary contribution, which is to show that SPE outperforms the only other fully-formulated method for stochastic supervised embeddings, the Hedged Instance Embedding (HIB). Our ‘toy-ish’ experiments are a superset of the complete set of tasks reported in the hedged-instance embedding (HIB) paper, which was presented at ICLR in 2019. SPE achieves superior performance over HIB across the board, and SPE has the additional benefit that training is more computation efficient (requiring only one sample per training example, vs. 8 for HIB). We also examine disentangling in SPE, which was not explored for HIB. Our interest in few-shot learning was mainly to show that—in the domain where Prototypical Networks (PNs) were developed—SPE performs as well and in some circumstances much better.  (Those circumstances are low-dimensional embeddings, and corrupted inputs.) We recognize that some researchers are (finally!) looking at larger data sets for few-shot learning, but Omniglot is one of two standard data sets used to justify PNs in the first place.  We have added additional comparisons of PNs and SPE (see Appendix). We have also rewritten our contributions sections (last para of Section 1, first para of Section 5) to emphasize our main focus.
>
> You say that the probabilistic formulation “does not seem to be new in terms of visual recognition”. Please provide us with citations.

---

> > ### Comment · AnonReviewer3 · 2019-11-14
> > **Related reference**
> >
> > I appreciate the response from the authors. I agree that SPE will have some advantages in few-shot learning tasks.
> >
> > Similar probabilistic formulation has been appeared in Bayesian face recognition and probabilistic k-means. Some papers have similar favor of formulation (just to name a few):
> >
> > -- Bayesian Face Revisited: A Joint Formulation, ECCV 2012
> > -- Bayesian Face Recognition, Vol. 33, No. 11, pps. 1771-1782, Pattern Recognition, 2000.
> > -- Surpassing Human-Level Face Verification Performance on LFW with GaussianFace, AAAI 2015
> > -- Object Perception as Bayesian Inference, Annu. Rev. Psychol., 2004
> > -- A Probabilistic Framework for Semi-Supervised Clustering, KDD 2004
> > -- Revisiting k-means: New Algorithms via Bayesian Nonparametrics, ArXiv 2012
> > -- Variational Prototyping-Encoder: One-Shot Learning with Prototypical Images, CVPR 2019
> >
> > I am not saying SPE has identical formulation with theirs, but the central ideas are quite similar, especially in those probabilistic clustering formulation.
> >
> > For the experiments, conducting few-shot recognition on Mini-ImageNet will strenghen the paper, and I will strongly suggest the authors consider to conduct experiments on these more realistic datasets. Omniglot is becoming saturated nowadays, and almost every new paper can get nearly perfect accuracy on it. I just don't know what may stop the authors from doing experiments on this dataset. Mini-ImageNet is not a large dataset, and it definitely does not take as much time as original ImageNet dataset.

---

> > > ### Author Response · Authors · 2019-11-15
> > > **Second Response to Review #3**
> > >
> > > Moghaddam et al. [2] and more specifically, Chen et al. [1], model the distribution of same-class and different-class pairs in a probabilistic manner. However,
> > >        * Embeddings are of input pairs, not single inputs.
> > >        * Embeddings are deterministic, not stochastic. There is no marginalization over uncertainty as in SPE.
> > >        * They estimate covariance statistics explicitly, rather than -- as in SPE -- learning a representation that has a certain covariance structure.
> > >
> > > Kersten et al. [4] is a highly general survey of object perception using Bayesian methods and is mostly outside the scope of SPE. The most-relevant concept is that of Bayesian information integration which is achieved via a product distribution. We also utilize a product distribution for integrating support instances into a prototype distribution.
> > >
> > > Basu et al. [5] propose a semi-supervised clustering method that utilizes pairwise constraints between instances and Kulis et al. [6] explore the relationship between classical clustering methods and Bayesian nonparametrics. The mathematics and model formulation don’t have many similarities to SPE, however, both suppose the existence of a cluster prototype, but that assumption is almost always made in clustering methods. If the implied analogy by the reviewer is that k-Means $\rightarrow$ GMM as PN $\rightarrow$ SPE, we find that comparison relatively weak. Namely, clustering methods are unsupervised (or semi-supervised in this case), and aren’t formulated as a recognition model. Furthermore, the notion that a GMM is a soft-clustering method doesn’t map directly to SPE, since SPE still assumes that a given instance belongs to exactly one class. There may be interesting relationships between [6] and [8]. Note that we acknowledge the existence of a combination of [8] and SPE in Section 5.
> > >
> > > Kim et al. [7] is most closely related to SPE, however, it is a generative model, not a recognition model. Furthermore, the generative component of [7] permits the use of variational inference techniques, which is foundationally different than our approach for training SPE.
> > >
> > > With regards to the referred articles, we do believe that SPE is a novel contribution, albeit with some minor similarities to prior work that use Bayesian techniques with shallow models for face verification. The most notable relationship is with [7] from CVPR 2019.
> > >
> > > [1] Dong Chen, Xudong Cao, Liwei Wang, Fang Wen, Jian Sun. “Bayesian Face Revisited: A Joint Formulation”. ECCV 2012.
> > > [2] Baback Moghaddam, Tony Jebara, Alex Pentland. “Bayesian Face Recognition”. Vol. 33, No. 11, pps. 1771-1782, Pattern Recognition, 2000.
> > > [3] Chaochao Lu, Xiaoou Tang. “Surpassing Human-Level Face Verification Performance on LFW with GaussianFace”. AAAI 2015.
> > > [4] Daniel Kersten, Pascal Mamassian, Alan Yuille. “Object Perception as Bayesian Inference”. Annu. Rev. Psychol., 2004.
> > > [5] Sugato Basu, Mikhail Bilenko, Raymond J. Mooney. “A Probabilistic Framework for Semi-Supervised Clustering”. KDD 2004.
> > > [6] Brian Kulis, Michael I. Jordan. “Revisiting k-means: New Algorithms via Bayesian Nonparametrics”. ICML 2012.
> > > [7] Junsik Kim, Tae-Hyun Oh, Seokju Lee, Fei Pan, In So Kweon. “Variational Prototyping-Encoder: One-Shot Learning with Prototypical Images”. CVPR 2019.
> > > [8] Kelsey R. Allen, Evan Shelhamer, Hanul Shin, Joshua B. Tenenbaum. “Infinite Mixture Prototypes for Few-Shot Learning”. ICML 2019.

---

### Decision · Program_Chairs · 2019-12-19

**Decision:**

Reject

**Comment:**

The consensus of reviewers is that this paper is not acceptable in present form, and the AC concurs.